# Height Estimation from Vertebral Parameters on Routine Computed Tomography in a Contemporary Elderly Australian Population: A Validation of Existing Regression Models

**DOI:** 10.3390/diagnostics13071222

**Published:** 2023-03-23

**Authors:** Damian Flanders, Timothy Lai, Numan Kutaiba

**Affiliations:** Department of Radiology, Austin Health, Melbourne, VIC 3084, Australia

**Keywords:** computed tomography, height estimation, artificial intelligence

## Abstract

The aim of this study is to compare previously published height estimation formulae in a contemporary Australian population using vertebral measurements readily available on abdominal CT. Retrospective analysis of patients undergoing a planning CT prior to transcatheter aortic valve implantation in a 12-month period was conducted; 96 participants were included in the analysis from a total of 137, with 41 excluded due to incomplete data. Seven vertebral measurements were taken from the CT images and height estimates were made for each participant using multiple regression equations from the published literature. Paired sample *t*-tests were used to compare actual height to estimated height. Many of the models failed to accurately predict patient height in this cohort, with only three equations for each sex resulting in a predicted height that was not statistically significantly different to actual height. The most accurate model in female participants was based on posterior sacral length and resulted in a mean difference between an actual and calculated height of 0.7 cm (±7.4) (*p* = 0.520). The most accurate model in male participants was based on anterior sacrococcygeal length and resulted in a mean difference of −0.6 ± 6.9 cm (*p* = 0.544). Height estimation formulae can be used to predict patient height from common vertebral parameters on readily available CT data. This is important for the calculation of anthropometric measures for a variety of uses in clinical medicine. However, more work is needed to generate accurate prediction models for specific populations.

## 1. Introduction

Morphometric measurements obtained from abdominal computed tomography (CT) have been shown to correlate with clinical outcomes, particularly in the surgical oncology and abdominal surgery populations [1,2]. These parameters can generally be measured from a single axial slice at the L3 level, including an assessment of sarcopaenia based on muscle cross-sectional area and density [3]. Fat compartments, particularly visceral fat, are associated with adverse metabolic outcomes, independent of standard anthropometric measurements like body mass index (BMI) and waist circumference [4]. The European Working Group on Sarcopenia in Older People (EWGSOP) published in 2019 a revised consensus definition of sarcopenia based on updated research [5]. Sarcopenia is now described as “a progressive and generalized skeletal muscle disorder that is associated with increased likelihood of adverse outcomes including falls, fractures, physical disability and mortality” [5]. Fat compartment measurements and sarcopenia are associated with adverse outcomes in a wide number of clinical conditions and provide valuable prognostic information beyond traditional risk factors [2,6].

Both muscle and fat compartment measurements from CT require standardization against patient body size parameters (e.g., height). However, a patient’s height is not routinely recorded during clinical assessment or on presentation for a radiological test. This lack of height measurement creates a missed opportunity for patients, many of whom have already undergone CT scanning during their clinical presentation, including for pre-operative planning [6]. An additional use for height and body size estimation is in the critical care setting, where height may be required for medication dosage calculations. For these reasons there have been various attempts made to estimate height from alternative available measurements and imaging [7,8,9,10,11,12,13]. Much of the literature regarding height estimation based on alternative measurements comes from the field of forensic medicine, where only parts of cadavers may be available following fires and mass disaster events [14]. Various methods for height estimation exist in this context, and the most accurate of these methods traditionally involve the long bones [8]. Methods have also been proposed for the estimation of height based on vertebral parameters on cross-sectional imaging [9,10,11,12,13]. The purpose of this study was to compare and validate previously published formulae for patient height prediction in a contemporary Australian population based on readily available vertebral measurements on abdominal CT examinations. To the knowledge of the authors, such methods for height estimation have not previously been applied to this population.

## 2. Materials and Methods

### 2.1. Cohort

This study was approved by the Office of Research at our institution and in keeping with the policies for a retrospective review, informed consent was not required. Patients who underwent a planning CT for elective trans-catheter aortic valve implantation (TAVI) procedures at our centre between 1 January and 31 December 2019 were reviewed. A total of 137 planning CT examinations were performed during this period. Of these, 18 were excluded due to lack of sufficient imaging field, such as exclusion of the lower sacrum. A further 23 examinations were excluded due to actual patient height data being unavailable. Complete CT and height data from a total of 96 participants were available for analysis.

### 2.2. Image Analysis and Measurements

All CT examinations were performed in the supine position and reviewed by one of the authors, with vertebral parameters measured manually on the institution’s picture archiving and communications system (IMPAX, Agfa, Mortsel, Belgium).

Seven vertebral measurements collected in our study are defined in Table A1 and displayed in Figure 1a,b. They include: vertebral body area (VBA), anteroposterior vertebral body diameter (APVBD), lateral vertebral body diameter (LVBD), anterior sacral length (ASL), posterior sacral length (PSL), anterior sacro-coccygeal length (ASCL) and posterior sacro-coccygeal length (PSCL). VBA, APVBD and LVBD were measured at both the 3rd and 4th lumbar vertebrae using the method previously outlined by Waduud et al. [10]. ASSL, PSSL, ASCL and PSCL were measured according to the method outlined by Torimitsu et al. [11]. Vertebral measurements were taken and recorded on an electronic worksheet by a trained investigator, supervised by an abdominal radiologist. Waduud et al. have previously demonstrated that trained individuals of varying clinical experience can accurately measure vertebral parameters [15]. The VBA measurement was recorded in squared centimetres (cm^2^) to the nearest decimal of a centimetre, and all linear vertebral parameters were recorded in millimetres (mm).

The measured vertebral parameters were then used to calculate heights in this cohort based on various formulae available in the literature for each sex (Table 1 and Table 2) [9,10,11,12,13]. Calculated heights were recorded to the nearest decimal of a centimetre (cm).

### 2.3. Data Collection

Actual patient height (cm) and demographics were extracted from the technologist worksheet completed at the time of scan acquisition. Height was recorded to the nearest centimetre on the technologist worksheet as per routine practice. Comorbidities were retrieved from the participants’ electronic medical records.

### 2.4. Statistical Analysis

Paired sample *t*-tests were used to compare the documented and predicted heights based on each regression equation. All data analyses were performed with IBM SPSS, version 22 (IBM Corp., Armonk, NY, USA). The mean difference between actual and predicted height, along with the standard deviation of this difference was recorded. A *p*-value of <0.05 was used as cut-off for statistical significance, with any value greater than this indicating that the means were not significantly different. The equation that produced the smallest mean difference for each sex was graphed on a Bland–Altman plot.

## 3. Results

Height and vertebral measurements were collected from 96 participants. There were 45 (47%) female and 51 (53%) male participants. Mean (±SD) documented height was 158.2 cm (±7.9) and 171.3 cm (±6.9), respectively. Descriptive statistics for age, height, weight and vertebral measurements are included in Table 3. Comorbidity distribution in the population is included in Table 4.

The predictive value of each of the published regression formulae is detailed in Table 5 and Table 6. The model with the closest height calculation to actual height in female participants was based on PSL and came from analysis of a contemporary Japanese population by Torimitsu et al. [11]. The calculated height based on this formula was 157.5 cm (±7.0) with a mean difference between actual and calculated height of 0.7 cm (±7.4) (*p* = 0.520). Two alternative formulae, both from the same Japanese study, were able to predict patient height within the accepted accuracy. A Bland–Altman plot for the most accurate equation in the female population is shown in Figure 2, comparing the difference between the predicted and actual height versus the average of the two measures.

The most accurate prediction model in male participants was also from the Japanese sample [11] and was based on ASCL, with a mean predicted height of 171.9 ± 2.7 cm and a mean difference of −0.6 ± 6.9 cm (*p* = 0.544). The majority of models analysed failed to predict patient height accurately in this population. Two alternative formulae were able to predict actual height with some accuracy in this population (based on a *p* > 0.05 for the difference between the mean of the actual vs. predicted height). Both relied on ASL, with one from the Japanese population described above and the other from an Anatolian Caucasian population. [11,13]. A Bland–Altman plot for the most accurate equation in the male population is shown in Figure 3, comparing the difference between the predicted and actual height versus the average of the two measures.

## 4. Discussion

Our study tested various vertebral measurements and formulae suggested by the available literature to calculate patients’ heights in an elderly Australian population who underwent planning CT studies for elective TAVI procedures. We found that calculated heights in our cohort were best derived from formulae obtained from a study on an elderly Japanese cadaveric population [11]. Patients undergoing planning CT scans for TAVI procedures were the focus of our study, as their actual heights were routinely recorded on the CT technologist worksheet and because this cohort represents an age group that is likely to benefit from assessment of sarcopenia and fat compartment measurements. The calculated heights can therefore allow standardized measurements of sarcopenia and fat compartments when actual heights are not available.

Our study showed that at least one model for each sex was able to provide height calculation with a potentially acceptable level of error. Formulae obtained from studies on other populations showed higher levels of error when applied to our cohort. Interestingly, the most accurate model for both male and female participants were from a study that relied on vertebral parameters measured on CT imaging of cadavers, rather than live patients, such as in the study by Oura et al. [12]. This was somewhat unexpected, as measured cadaver height differs from living height, likely due to tissue changes that occur after death [16]. One factor that may account for this is the variation in modality used for imaging. Oura et al. [12] used Magnetic Resonance Imaging (MRI) of the lumbar spine, while our measures, and all other formula in this validation, were based on CT measurements. Age plays an important role in height [17]. The advanced age of this cohort differs from many of the studies from which the estimation formulae were taken, likely reducing their accuracy. Diurnal variation also has an appreciable impact on patient height [17]. Due to logistical reasons and the retrospective nature of the study we did not record or specify the time of the day when measurements were taken, as was done in Zhang et al. [18].

Previous studies have consistently found equations from multiple linear regression analysis to be the most accurate [7,14,18]. This study looked at single-parameter models in order to determine if the accuracy of these simpler methods was adequate, as the measurement of a single vertebral parameter would not take a large amount of time for a reporting radiologist. However, increasing numbers and complexity of these measurements could make height estimation impractical for everyday use. The advancing capability of artificial intelligence raises the possibility of built-in algorithms for the measurement of vertebral parameters and automatic estimation of height based on highly complex formulae. Further research into the most accurate model of estimation is therefore warranted, regardless of computational complexity.

Direct measurement of actual height remains the preferred method for standardization of muscle and fat compartment measurements obtained from CT examinations. However, large numbers of patients who undergo abdominal imaging lack documentation of actual height in their medical records. The ability to calculate patient height from available cross-sectional imaging potentially allows for retrospective analysis of large numbers of CT scans even when actual heights are not documented. This approach becomes particularly relevant with progress in artificial intelligence algorithms performing rapid calculations of cross-sectional areas of muscle and fat compartments from a single slice CT image [19]. Measurement of parameters such as psoas muscle cross-sectional area and total body cross-sectional area on CT have been shown to correlate with important clinical factors such as total body mass and visceral fat mass [20]. Automation of this process could therefore provide clinically useful information without effort from the reporting radiologist in the future.

There were several limitations to our study. Firstly, the method for measurement of actual heights of our patients was not specified due to the retrospective nature of this analysis. Patients undergoing planning CT for TAVI procedures routinely have their height recorded by the technologists overseeing the scan. This could have been done by direct measurement against stepped height labels marked on the wall of the CT room, a stadiometer or self-reported by the patient. Unfortunately, people tend to over-estimate their own height on self-reporting [21]. Secondly, our sample size was small and consisted of elderly patients being prepared for TAVI procedures. The mean age in female participants was 81.8 years and in male participants was 79.1 years. This compares to 56.4 years and 56.7 years in the study by Torimitsu et al. [11]. The age group of our cohort is the focus of existing and ongoing research in interventions addressing sarcopenia and is therefore a useful target population for attempts at calculating such parameters [22]. Our elderly cohort was chosen to validate existing formulae rather than to derive a new formula for height calculation. As such, a larger sample size may not be needed. The importance of population-specific regression models for accurate height estimation is well recognised in the literature [9,12,13]. This includes the need for further investigation into the accuracy of models for different age groups even within a population of specific ancestry [23]. The authors are not aware of any existing models in the contemporary Australian population, and this study did not have the scope to generate and test its own formula. By validating existing formulae from other populations in our cohort, we could potentially use such formulae in similar Australian cohorts with existing CT studies but no documented height. Thirdly, we did not record the ethnicity of the patients in our cohort, as some of this information is incomplete or not accurately documented. Australia’s ethnic diversity presents a challenge for the application of estimation models due to the greater anthropometric variation that migration and multinational mobility creates [24]. Calculated models may not apply to the indigenous Australian population specifically, and this requires further research. Finally, patients with metabolic or other pathologies that may affect vertebral parameters, such as osteoarthritis and osteoporosis, were not excluded from this study, as was the case in much of the literature [12,18]. Thirteen percent of female participants and 2% of male participants had a documented diagnosis of osteoporosis in their medical record, which undoubtedly would reduce the accuracy of the estimations. However, such conditions are ubiquitous in elderly populations.

## 5. Conclusions

Anthropometric measurements of patients using CT imaging has the potential to improve risk stratification in a variety of contexts. The rapid identification of incidental sarcopaenia or visceral fat indexes on routine scanning could be used to identify individuals that would benefit from targeted intervention to reverse such diagnoses. Height estimation plays a vital role in these calculations, and this study shows that some of the available formulae based on readily measurable vertebral parameters can be used in new contexts to provide acceptable levels of accuracy.

## Figures and Tables

**Figure 1 diagnostics-13-01222-f001:**
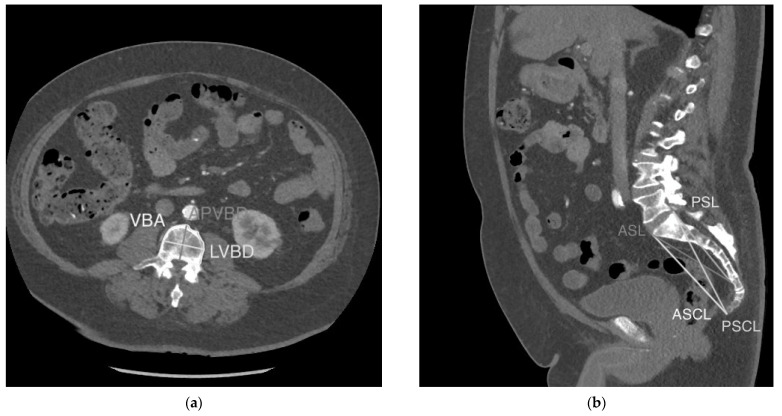
Vertebral measurements. (**a**). VBA, LVBD and APVBD displayed on an axial section through the superior endplate of L3. (**b**). ASL, PSL, ASCL and PSCL displayed on a mid-sagittal section through the abdomen/pelvis.

**Figure 2 diagnostics-13-01222-f002:**
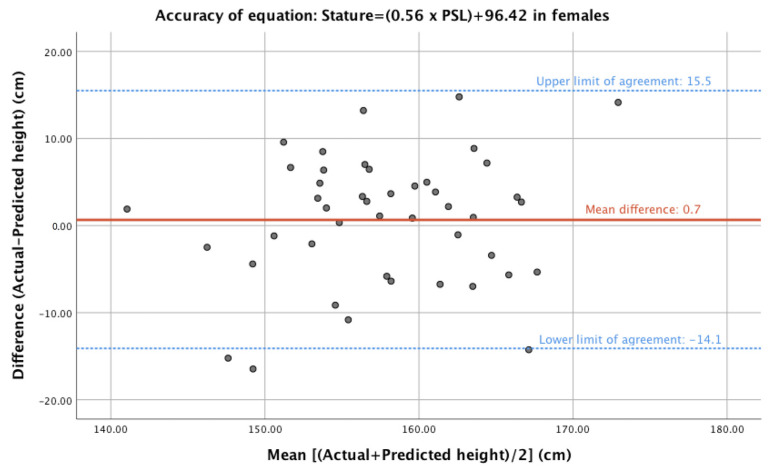
Bland–Altman plot of difference (actual—predicted height) and mean (of actual and predicted heights) for the most accurate model for female participants.

**Figure 3 diagnostics-13-01222-f003:**
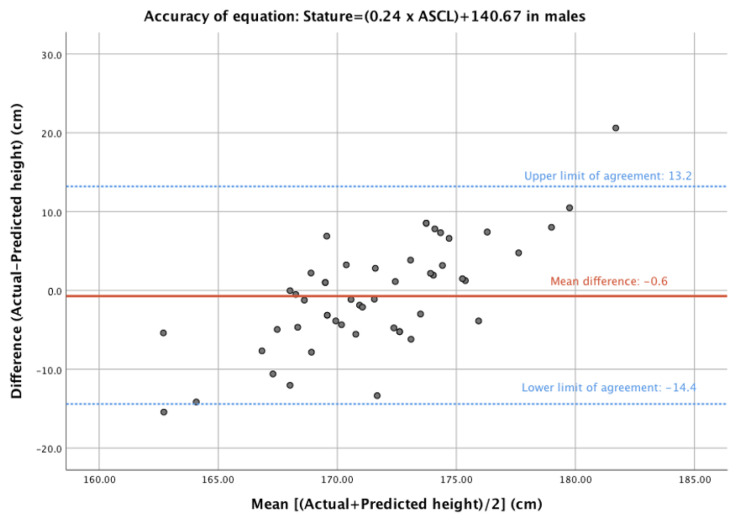
Bland–Altman plot of difference (actual—predicted height) and mean (of actual and predicted heights) for the most accurate model for male participants.

**Table 1 diagnostics-13-01222-t001:** Regression equations used to generate height estimates for female participants.

Study	Age (Mean +/− SD)	Vertebral Parameter Used	Regression Equation (H = Height (cm))	Accuracy
**Zhan et al. (2018)** [9] Contemporary Chinese population (n = 160)	50 ± 13	ASL	$H=133.867+\left[1.948\times \mathrm{ASL}\left(\mathrm{cm}\right)\right]$	SEE = 5.546
		PSL	$H=121.095+\left[3.138\times \mathrm{PSL}\left(\mathrm{cm}\right)\right]$	SEE = 5.268
		ASCL	$H=135.599+\left[1.599\times \mathrm{ASCL}\left(\mathrm{cm}\right)\right]$	SEE = 5.525
		PSCL	$H=122.762+\left[2.525\times \mathrm{PSCL}\left(\mathrm{cm}\right)\right]$	SEE = 5.239
**Torimitsu et al. (2014)** [11] Contemporary Japanese cadaveric population (n = 106)	56 ± 20	ASL	$H=\left(0.38\times \mathrm{ASL}\left(\mathrm{mm}\right)\right)+115.33$	SEE = 7.84 cm
		PSL	$H=\left(0.56\times \mathrm{PSL}\left(\mathrm{mm}\right)\right)+96.42$	SEE = 7.15 cm
		ASCL	$H=\left(0.47\times \mathrm{ASCL}\left(\mathrm{mm}\right)\right)+98.93$	SEE = 7.41 cm
		PSCL	$H=\left(0.56\times \mathrm{PSCL}\left(\mathrm{mm}\right)\right)+85.29$	SEE = 6.68 cm
**Oura et al. (2018)** [12] Contemporary middle-aged Finnish population (n = 742)	47 ± 0	VBA	$H=\left[2.314\times \mathrm{CSA}\left({\mathrm{cm}}^{2}\right)\right]+140.331$	SEE = 4.971 cm
**Waduud et al. (2019)** [10] Contemporary elderly British population (n = 255, combined sex cohort)	75 ± 8 (combined sex cohort)	L3 VBA	$H=157.4+\left[1.3\times \mathrm{VBA}\left({\mathrm{cm}}^{2}\right)\right]-[\mathrm{age}\left(\mathrm{years}\right)\times 0.1$	Difference from actual height (mean ± SD): 0.6 ± 6.3 (*p* = 0.276)
		L3 APVBD	$H=143.1+\left[\mathrm{APVBD}\left(\mathrm{cm}\right)\times 9.1]-[\mathrm{age}\left(\mathrm{years}\right)\times 0.1\right]$	Difference from actual height (mean ± SD): 0.8 ± 6.6 (*p* = 0.164)
		L3 LVBD	$H=149.2+\left[\mathrm{LVBD}\left(\mathrm{cm}\right)\times 5.7]-[\mathrm{age}\left(\mathrm{years}\right)\times 0.1\right]$	Difference from actual height (mean ± SD): 0.7 ± 6.6 (0.222)

**Table 2 diagnostics-13-01222-t002:** Regression equations used to generate height estimates for male participants.

Study	Age (Mean ± SD)	Vertebral Parameter Used	Regression Equation (H = Height (cm))	Accuracy
**Zhan et al. (2018)** [9] Contemporary Chinese population (n = 190)	55 ± 11	ASL	$H=136.477+\left[2.662\times \mathrm{ASL}\left(\mathrm{cm}\right)\right]$	SEE = 5.829
		PSL	$H=130.756+\left[3.126\times \mathrm{PSL}\left(\mathrm{cm}\right)\right]$	SEE = 5.716 cm
		ASCL	$H=145.507+\left[1.619\times \mathrm{ASCL}\left(\mathrm{cm}\right)\right]$	SEE = 6.094
		PSCL	$H=139.003+\left[1.989\times \mathrm{PSCL}\left(\mathrm{cm}\right)\right]$	SEE = 6.012
**Torimitsu et al. (2014)** [11] Contemporary Japanese cadaveric population (n = 110)	57 ± 20	ASL	$H=\left(0.32\times \mathrm{ASL}\left(\mathrm{mm}\right)\right)+132.69$	SEE = 5.98 cm
		PSL	$H=\left(0.43\times \mathrm{PSL}\left(\mathrm{mm}\right)\right)+143.67$	SEE = 5.83 cm
		ASCL	$H=\left(0.24\times \mathrm{ASCL}\left(\mathrm{mm}\right)\right)+140.67$	SEE = 5.94 cm
		PSCL	$H=\left(0.42\times \mathrm{PSCL}\left(\mathrm{mm}\right)\right)+154.81$	SEE = 5.98 cm
**Karakas et al. (2011)** [13] Contemporary Anatolian Caucasian population (n = 66)	42 ± 15	ASL	$H=\left(0.306\times \mathrm{ASL}\left(\mathrm{mm}\right)\right)+137.9\;$	SEE = 5.69 cm
**Oura et al. (2018)** [12] Contemporary middle-aged Finnish population (n = 616)	47 ± 0	L4 VBA	$H=\left[1.702\times \mathrm{CSA}\left({\mathrm{cm}}^{2}\right)\right]+156.116$	SEE = 5.412 cm
**Waduud et al. (2019)** [10] Contemporary elderly British population (n = 255, combined sex cohort)	75 ± 8 (combined sex cohort)	L3 VBA	$H=157.4+\left[1.3\times \mathrm{VBA}\left({\mathrm{cm}}^{2}\right)\right]-\left[\mathrm{age}\left(\mathrm{years}\right)\times 0.1\right]+6.7$	Difference from actual height (mean ± SD): 0.6 ± 6.3 (*p* = 0.276)
		L3 APVBD	$H=143.1+\left[\mathrm{APVBD}\left(\mathrm{cm}\right)\times 9.1]-[\mathrm{age}\left(\mathrm{years}\right)\times 0.1\right]+6.8$	Difference from actual height (mean ± SD): 0.8 ± 6.6 (*p* = 0.164)
		L3 LVBD	$H=149.2+\left[\mathrm{LVBD}\left(\mathrm{cm}\right)\times 5.7]-[\mathrm{age}\left(\mathrm{years}\right)\times 0.1\right]+7.9$	Difference from actual height (mean ± SD): 0.7 ± 6.6 (0.222)

**Table 3 diagnostics-13-01222-t003:** Descriptive statistics of all variables.

	Female Participants, n = 45				Male Participants, n = 51			
	Minimum	Maximum	Mean	Std. Deviation	Minimum	Maximum	Mean	Std. Deviation
**Age (years)**	62.0	92.0	81.8	±6.2	57.0	92.0	79.1	±7.7
**Height (cm)**	140.0	180.0	158.2	±7.9	155.0	192.0	171.3	±6.9
**Weight (kg)**	40	143	68.9	21.7	49	146	81.6	20
**L3 VBA**	9.9	16.0	12.8	±1.6	12.5	22.5	16.0	±2.3
**L3 LVBD**	32.5	52.6	43.9	±4.4	41.6	59.9	49.2	±3.7
**L3 APVBD**	30	46.5	33.3	±3.1	31.8	46.5	37.0	±3.2
**L4 VBA**	9.9	16.1	13.5	±1.7	12.2	24.3	16.8	±2.6
**L4 LVBD**	36.5	55.5	45.6	±4.1	43.3	72.8	50.7	±4.8
**L4 APVBD**	28.6	38.0	33.1	±2.0	31.8	45.1	36.9	±3.1
**ASSL**	69.5	133.0	108.0	±13.5	98.0	147.0	115.8	±10.8
**PSSL**	78.0	139.0	109.1	±12.4	100.0	146.0	119.5	±10.4
**ASCL**	86.4	152.0	118.5	±15.3	103	157	130.0	±11.4
**PSCL**	98.4	157	126.4	±14.4	118	165	141.9	±10.3

**Table 4 diagnostics-13-01222-t004:** Comorbidities by sex (Female, n = 45) (Male, n = 51).

Comorbidity	Female Sex, Number (%)	Male Sex, Number (%)
**Osteoporosis**	6 (13)	1 (2)
**Hypertension**	34 (78)	45 (88)
**Ischaemic heart disease**	10 (22)	27 (53)
**Cerebrovascular disease**	7 (16)	10 (20)
**Diabetes mellitus**	12 (27)	20 (39)
**Chronic kidney disease**	11 (24)	12 (24)
**Chronic liver disease**	2 (4)	2 (4)
**Cancer**	11 (24)	9 (18)

**Table 5 diagnostics-13-01222-t005:** Accuracy of regression formulae for height calculation in female participants.

Study	Regression Equation	Calculated Height (Mean ± SD)	Difference from Actual Height (Mean ± SD)	*p*-Value
**Zhan et al., 2018** [9]	$\mathrm{Height}\left(H\right)\mathrm{in}\;\mathrm{cm}=133.867+\left[1.948\times \mathrm{ASL}\left(\mathrm{cm}\right)\right]$	154.9 ± 2.6	3.3 ± 6.7	0.002
**Zhan et al., 2018** [9]	$H=121.095+\left[3.138\times \mathrm{PSL}\left(\mathrm{cm}\right)\right]$	155.3 ± 3.9	2.9 ± 6.8	0.007
**Zhan et al., 2018** [9]	$H=135.599+\left[1.599\times \mathrm{ASCL}\left(\mathrm{cm}\right)\right]$	154.5 ± 2.5	3.7 ± 7.0	0.001
**Zhan et al., 2018** [9]	$H=122.762+\left[2.525\times \mathrm{PSCL}\left(\mathrm{cm}\right)\right]$	154.7 ± 3.6	3.5 ± 7.1	0.002
**Torimitsu et al., 2014** [11]	$H=\left(0.38\times \mathrm{ASL}\left(\mathrm{mm}\right)\right)+115.33$	156.4 ± 5.2	1.8 ± 6.5	0.063
**Torimitsu et al., 2014** [11]	$H=\left(0.56\times \mathrm{PSL}\left(\mathrm{mm}\right)\right)+96.42$	157.5 ± 7.0	0.7 ± 7.4	0.523
**Torimitsu et al., 2014** [11]	$H=\left(0.47\times \mathrm{ASCL}\left(\mathrm{mm}\right)\right)+98.93$	154.6 ± 7.2	3.6 ± 7.5	0.003
**Torimitsu et al., 2014** [11]	$H=\left(0.56\times \mathrm{PSCL}\left(\mathrm{mm}\right)\right)+85.29$	156.1 ± 8.0	2.1 ± 8.4	0.100
**Oura et al., 2018** [12]	$H=\left[2.314\times \mathrm{CSA}\left({\mathrm{cm}}^{2}\right)\right]+140.331$	171.6 ± 3.8	−13.4 ± 7.3	0.000
**Waduud et al., 2019** [10]	$H=157.4+\left[1.3\times \mathrm{VBA}\left({\mathrm{cm}}^{2}\right)\right]-\left[\mathrm{age}\left(\mathrm{years}\right)\times 0.1\right]$	165.9 ± 2.0	−7.7 ± 7.6	<0.001
**Waduud et al., 2019** [10]	$H=143.1+\left[\mathrm{APVBD}\left(\mathrm{cm}\right)\times 9.1]-[\mathrm{age}\left(\mathrm{years}\right)\times 0.1\right]$	165.3 ± 2.8	−7.1 ± 8.3	0.000
**Waduud et al., 2019** [10]	$H=149.2+\left[\mathrm{LVBD}\left(\mathrm{cm}\right)\times 5.7]-[\mathrm{age}\left(\mathrm{years}\right)\times 0.1\right]$	166.0 ± 2.5	−7.8 ± 7.6	0.000

**Table 6 diagnostics-13-01222-t006:** Accuracy of formulae equations for height calculation in male participants.

Study	Regression Equation	Calculated Height (Mean ± SD)	Difference from Actual Height (Mean ± SD)	*p*-Value
**Zhan et al., 2018** [9]	$\mathrm{Height}\left(H\right)\mathrm{in}\;\mathrm{cm}=136.477+\left[2.662\times \mathrm{ASL}\left(\mathrm{cm}\right)\right]$	167.3 ± 2.9	4.0 ± 7.2	0.000
**Zhan et al., 2018** [9]	$H=130.756+\left[3.126\times \mathrm{PSL}\left(\mathrm{cm}\right)\right]$	168.1 ± 3.3	3.2 ± 7.0	0.002
**Zhan et al., 2018** [9]	$H=145.507+\left[1.619\times \mathrm{ASCL}\left(\mathrm{cm}\right)\right]$	166.6 ± 1.9	4.7 ± 6.8	0.000
**Zhan et al., 2018** [9]	$H=139.003+\left[1.989\times \mathrm{PSCL}\left(\mathrm{cm}\right)\right]$	167.2 ± 2.0	4.1 ± 6.5	0.000
**Torimitsu et al., 2014** [11]	$H=\left(0.32\times \mathrm{ASL}\left(\mathrm{mm}\right)\right)+132.69$	169.7 ± 3.4	1.5 ± 7.4	0.145
**Torimitsu et al., 2014** [11]	$H=\left(0.43\times \mathrm{PSL}\left(\mathrm{mm}\right)\right)+143.67$	195.0 ± 4.5	−23.8 ± 7.4	0.000
**Torimitsu et al., 2014** [11]	$H=\left(0.24\times \mathrm{ASCL}\left(\mathrm{mm}\right)\right)+140.67$	171.9 ± 2.7	−0.6 ± 6.9	0.544
**Torimitsu et al., 2014** [11]	$H=\left(0.42\times \mathrm{PSCL}\left(\mathrm{mm}\right)\right)+154.81$	214.4 ± 4.3	−43.1 ± 6.9	0.000
**Karakas et al., 2011** [13]	$H=\left(0.306\times \mathrm{ASL}\left(\mathrm{mm}\right)\right)+137.9$	173.3 ± 3.3	−2.1 ± 7.3	0.051
**Oura et al., 2018** [12]	$H=\left[1.702\times \mathrm{CSA}\left({\mathrm{cm}}^{2}\right)\right]+156.116$	184.7 ± 4.4	−13.4 ± 7.6	0.000
**Waduud et al., 2019** [10]	$H=157.4+\left[1.3\times \mathrm{VBA}\left({\mathrm{cm}}^{2}\right)\right]-\left[\mathrm{age}\left(\mathrm{years}\right)\times 0.1\right]+6.7$	177.0 ± 3.0	−5.74 ± 6.9	0.000
**Waduud et al., 2019** [10]	$H=143.1+\left[\mathrm{APVBD}\left(\mathrm{cm}\right)\times 9.1]-[\mathrm{age}\left(\mathrm{years}\right)\times 0.1\right]+6.8$	175.7 ± 3.1	−4.4 ± 6.4	0.000
**Waduud et al., 2019** [10]	$H=149.2+\left[\mathrm{LVBD}\left(\mathrm{cm}\right)\times 5.7]-[\mathrm{age}\left(\mathrm{years}\right)\times 0.1\right]+7.9$	177.2 ± 2.1	−6.0 ± 6.9	0.000

## Data Availability

The data presented in this study are available on request from the corresponding author.

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
