# Peer review of "Height Estimation from Vertebral Parameters on Routine Computed Tomography in a Contemporary Elderly Australian Population: A Validation of Existing Regression Models"

_diagnostics, 2023, doi:10.3390/diagnostics13071222_

Round 1
Reviewer 1 Report
Dear authors, I must say that it was an interesting article to read. However, prior the publication, there are some issues that can be resolved.
Introduction
Although, there is some explanation on why there is a need for the indirect height assessment, some other possible reasons can be added as well. Is it only for correcting the formulas for sarcopenia and fat compartment measurements?
Methods:
Is there a number or a date of ethical approval? Please add.
Line 64, is it CT study or CT examination/ scans?
Line 67, total of 96 participants, instead of subjects, subjects is more appropriate for the studies done on the laboratory animals, not humans. Please replace ‘subject’ with ‘participant’ throughout the manuscript.
Also, I am a bit confused with the choice of the study population. How does this population differ from the general population? Is there any data comparing this and the general population?
Results
Also, use ‘female participants’ and ‘male participants’ instead of ‘females’ and ‘males’.
Discussion
As said before, I would like to see it as a part of discussion, the explanation on the choice of the population? Pros and cons? How it may have impacted the results?
Line 172, I think it is ‘sex’ instead of ‘gender’, as this is more of biological than social examination.
Author Response
Dear reviewer,
Thank you for your considered input regarding our paper. We found your suggestions incredibly useful and have revised the paper to take them into account, which we think has improved both its content and structure.
Specifically:
- We have included the additional use of calculated height for the use in emergency drug calculations when height is not immediately available.
- The ethics committee details are stated following the conclusion section- before the references.
- CT "study" has been replaced with "examination" where relevant.
- "Subjects" has been replaced with "participants" throughout the paper.
- "Participants" has been added before female and male where relevant.
- We have added some additional details regarding the age of our population (older mean age than the previous literature) and why this is useful (the elderly are the target population for interventions to address pre-operative sarcopenia).
- "Gender" has been replaced with "sex" where relevant.
Thank you again for your consideration.
Kind regards,
Damian, Numan and Tim.
Reviewer 2 Report
After reviewing this manuscript entitled (Height estimation from vertebral parameters on routine computed tomography in a contemporary elderly Australian population: a validation of existing regression models), I found it an interesting work. However, there are some remarks have to be considered before recommending it for publication.
1- Please remove the full stop from the title.
2- Some complex sentences arise through the manuscript. Shorter and more comprehensive ones will be better.
3- The novelty of this work has to be expressed in the introduction section clearly stating the new achievement when compared to published corresponding studies.
4- The conclusion section is missing. It has to be written clearly.
5- Recent relevant references are recommended to be cited. The most recent one was published in 2020.
Author Response
Dear reviewer,
Thank you for your considered input regarding our paper. We found your suggestions incredibly useful and have revised the paper to take them into account, which we think has improved both its content and structure.
Specifically:
- The full stop has been removed from the title.
- Some changes to sentence structure have been made in an attempt to make it more clear.
- Addition of some information in the introduction regarding the novelty of this paper- application of height estimation formulae to this particular population.
- A concise conclusion has been placed at the end of the article.
- An additional 2 references from 2020 have been added- there have been no relevant articles from the past 24 months identified.
Thank you again for your consideration.
Kind regards,
The authors.
Round 2
Reviewer 1 Report
I would like to thank the authors for the corrections made. It is good article now.